# Octopamine Rescues Endurance and Climbing Speed in *Drosophila Clk^out^* Mutants with Circadian Rhythm Disruption

**DOI:** 10.3390/cells12212515

**Published:** 2023-10-25

**Authors:** Maryam Safdar, Robert J. Wessells

**Affiliations:** Department of Physiology, School of Medicine, Wayne State University, Detroit, MI 48201, USA; maryam.safdar@med.wayne.edu

**Keywords:** *Drosophila*, circadian disruption, endurance exercise, octopamine

## Abstract

Circadian rhythm disturbances are associated with various negative health outcomes, including an increasing incidence of chronic diseases with high societal costs. While exercise can protect against the negative effects of rhythm disruption, it is not available to all those impacted by sleep disruptions, in part because sleep disruption itself reduces exercise capacity. Thus, there is a need for therapeutics that bring the benefits of exercise to this population. Here, we investigate the relationship between exercise and circadian disturbances using a well-established *Drosophila* model of circadian rhythm loss, the *Clk^out^* mutant. We find that *Clk^out^* causes reduced exercise capacity, measured as post-training endurance, flight performance, and climbing speed, and these phenotypes are not rescued by chronic exercise training. However, exogenous administration of a molecule known to mediate the effects of chronic exercise, octopamine (OA), was able to effectively rescue mutant exercise performance, including the upregulation of other known exercise-mediating transcripts, without restoring the circadian rhythms of mutants. This work points the way toward the discovery of novel therapeutics that can restore exercise capacity in patients with rhythm disruption.

## 1. Introduction

Many biological processes in living organisms display rhythmic changes over the course of a 24 h day which are referred to as circadian rhythms [1,2,3]. Rhythmicity can be found in both behaviors and metabolism, including sleep cycles, hormonal changes, blood pressure, and body temperature [1,2,3]. These circadian rhythms are controlled by molecular clocks [4,5] that consist of self-sustaining transcriptional translational feedback loops or TTFLs [6]. These feedback loops can be entrained to several external stimuli, like light and temperature; however, they can continue to cycle on an intrinsic rhythm without external stimuli as well [7]. The feedback loops and their functions are preserved across many species, from mammals to invertebrates like *Drosophila* [4,5]. The master regulator, often referred to as the central clock, is located in the central nervous system [6]. The central clock consists of one main feedback loop involving four genes [6]. In *Drosophila*, these include the genes that make up the positive limb of the loop, *clock* and *cycle*, and those that make up the negative limb, *period* and *timeless* [4,7]. Clock and Cycle form a heterodimer and enter the nucleus to increase the expression of Period and Timeless [7]. These proteins also form a heterodimer once produced and inhibit the function of Clock and Cycle [7]. Once Timeless is degraded, in the presence of light, the inhibition on Clock and Cycle is removed and the cycle can begin again with the transcription and translation of *period* and *timeless* [7]. Disruption in any of the four genes can lead to a disruption of the cycle and loss of the central circadian rhythm [6,8]. The central clock controls rhythmic activities in part by communicating with and modulating peripheral clocks in other tissues, including muscle [4,5,9]. 

The disruption of circadian rhythms can occur due to various causes, including disease, shift work, jet lag, or old age [10,11,12], and is associated with diverse negative health outcomes. In mammals, rhythm disturbances are associated with metabolic syndrome, type 2 diabetes, obesity, reduced cardiovascular health, and overall mortality [12,13,14,15,16]. Likewise, *Drosophila* experience negative impacts on neurological health and health span with circadian rhythm disturbance [17,18]. Circadian mutants, carrying mutations in one of the core clock genes, display a sensitivity to oxidative stress, and rhythm disruption has also been found to be associated with increased neurodegeneration in flies [17,18,19,20]. 

Conversely, regular exercise is associated with improved cardiovascular, metabolic, and neurological health both in general [21,22], and specifically in the context of rhythm disruption [23,24,25]. For example, mice missing VIP (vasoactive intestinal peptide), a neuropeptide produced in the suprachiasmatic nucleus that is important for entrainment of circadian rhythms, have altered rhythms of activity, heart rate, and body temperature [26,27]. VIP-deficient mice were shown to regain lost rhythms of body temperature, heart rate, and activity with 2 weeks of voluntary exercise during the active phase of the nocturnal mammal [6,27]. 

However, the potential benefits of endurance exercise in the context of circadian rhythm disturbance may be limited by the impact of circadian disturbance on exercise capacity itself [28,29,30]. Loss-of-function mutations in some central clock proteins lead to reduced exercise capacity [28]. The overexpression of central clock protein inhibitors can also cause reduced exercise capacity in mice [29]. Furthermore, environmental factors that cause circadian disturbance, such as shift work, may concurrently reduce access to optimal or regular exercise [12]. Mammalian and human studies have shown the importance of time-of-day for exercise, and schedules like those of shift workers may not allow for optimized training [12,30,31,32,33]. 

Thus, further investigation into the interactions between endurance exercise and circadian rhythm disruption is required to discover methods of bypassing these limitations. *Drosophila* is a useful model for these studies, as methods to manipulate and measure both circadian rhythms and endurance exercise have been well established [4,23,34,35]. Here, we use the *Clk^out^* mutant as a model of rhythm disruption in *Drosophila* to investigate interactions between circadian rhythms and endurance exercise [8,36].

We have previously identified several key pathways and mediators required for endurance exercise to produce adaptive benefits in *Drosophila* [34], and shown that these mediators can mimic the effects of endurance exercise training in sedentary flies [37,38,39]. One example is the biogenic amine octopamine [38]. Octopamine (OA) is the functional equivalent of norepinephrine in invertebrates, and its release from octopaminergic neurons and binding to receptors is required for exercise training to produce benefits [38,40]. Feeding of OA to adult flies is sufficient to mimic the effects of training on endurance and climbing speed in *Drosophila* [38]. Hence, OA is both necessary and sufficient to coordinate the benefits of regular endurance exercise in *Drosophila*. 

Here, we describe a reduced exercise capacity in *Clk^out^* mutants. We find that OA feeding can substantially rescue endurance and speed of *Clk^out^* mutants without rescuing their circadian rhythms. Further, we go on to identify multiple downstream targets of OA in this process.

## 2. Materials and Methods

### 2.1. Fly Stocks and Maintenance

All flies were raised in 25 °C incubators with 50% humidity, on a 10% yeast/sugar diet unless otherwise specified. All flies were also maintained in a 12 h light:dark cycle, excluding those used for assessment of activity rhythm within the standard *Drosophila* activity monitor setup. All fly lines used were obtained from the Bloomington *Drosophila* Stock Center; *w*^1118^: BDSC3605; *Clk^out^*: BDSC56754 (Bloomington Stock Center, Bloomington, IN, USA). The *Clk^out^* mutants have previously been described as having an amorphic mutation in the *Clk* gene and as such have disrupted rhythms of activity and sleep, as they do not produce one of the key proteins involved in circadian cycling [8]. All flies were collected within 48–72 h post-eclosion and cohorts collected within this time frame were considered to be age-matched. 

### 2.2. Exercise

Flies were exercised using an involuntary (i.e., forced) exercise program [41]. Cohorts of at least 1000 flies were collected within 48–72 h of eclosion and separated into vials of 20 flies each. These flies were then further separated into groups of 500 flies each, the exercised (Ex) and unexercised (Un) groups. The vials were topped with foam stoppers and both groups were housed within the conditions described above. The flies were exercised using the Power Tower apparatus (induces automated negative geotaxis response) as previously described [41], with a ramped exercise program of gradually increasing daily duration for three weeks (1.5 h/day first week, 2 h/day second week, 2.5 h/day third week.). Exercise took place five times/week with two rest days each week. Unexercised flies were also placed on the machine at the time of training but were prevented from exercising by having the foam stopper pushed to the bottom of the vial. Assessments were conducted 3–4 days after conclusion of training [41]. All exercise was started around 2 h after lights on for all groups.

### 2.3. Octopamine Feeding 

Flies were transferred, at 2 days after eclosion, to food containing 5 μg/mL of octopamine (obtained from Sigma-Aldrich, St. Louis, MO, USA) or ddH_2_O vehicle. Experimental flies were maintained on food containing OA whereas controls were maintained on food containing the vehicle. All flies were housed in the same incubator. The concentration of octopamine used was based on previously published dose-response experiments [38].

### 2.4. Endurance 

Endurance was measured by using the Power Tower exercise apparatus to assess time-to-fatigue as previously described [42]. Eight vials containing 20 flies each were placed on the Power Tower and stimulated to climb until exhaustion. Vials were observed at 15 min intervals. A vial was considered fatigued when at least 80 percent of the flies within were unable to climb above 1 cm for four consecutive drops. At this point the vial was removed, and the time of removal was recorded in minutes. GraphPad Prism (Version 10.0.0, GraphPad, San Diego, CA, USA) was used to generate survival curves of the data and significance was assessed by a log-rank test.

### 2.5. Climbing Speed Evaluation

Climbing speed was evaluated using a modified version of the Rapid Negative Geotaxis (RING) assay as previously described [41,43]. Briefly, cohorts of 100 flies, housed in vials of 20 flies each, were transferred to empty vials mounted in the RING apparatus. After allowing 1 min for assimilation to the new environment, the vials were briskly tapped down, and a picture was taken 2 s after the last tap. Four consecutive trials were conducted for each group daily for 5 days a week over the course of the three-week-long experiments. All experiments were conducted approximately 1–2 h after lights on and prior to the start of the exercise program for the day. The images were batch processed using ImageJ (Version 1.54d, NIH and LOCI, Bethesda, MD, USA) and the data were graphed and analyzed using GraphPad Prism (Version 10.0.0, GraphPad, San Diego, CA, USA). A two-way ANOVA (to account for effects of genotype or treatment, and age) with Tukey’s post-hoc multiple comparisons was used to assess statistical differences.

### 2.6. Flight Performance Test

A cohort of eight vials, with 20 flies each, was assessed from each group as previously described [41]. Briefly, a Tangle-Trap (sticky fly trapping material, Olson Products Inc., Medina, OH, USA) coated polycarbonate sheet was rolled up and placed within a large cast acrylic tube. The acrylic tube was held in place with a large sized ring stand, the ring attachment was replaced with two chains near the top and bottom of the stand to hold the large tube in place. A plastic funnel with a plastic drop tube affixed to it was centered above the acrylic tube, by use of a smaller ring stand. Vials of flies with the foam stopper removed were dropped down the drop tube, one at a time. The drop down stimulated the flies to fall out of the vials and into the large acrylic tube lined with the sticky polycarbonate sheet. The sheet was removed and propped up next to a meter stick for scale and photographed. Flies stuck in the Tangle-Trap were removed and disposed of after being photographed and the Tangle-Trap was smoothed out before the polycarbonate sheet was returned to the acrylic tube for testing of the next group. Images were processed using ImageJ (Version 1.54d, NIH and LOCI, Bethesda, MD, USA) and data was analyzed using GraphPad Prism (Version 10.0.0, GraphPad, San Diego, CA, USA). A one-way ANOVA or *t*-test, depending on the number of groups, was used to evaluate statistically significant differences.

### 2.7. Activity Rhythm Evaluation

The Drosophila Activity Monitors (DAM) (TriKinetics, Waltham, MA, USA) were used to evaluate activity rhythms on cohorts of 16 flies from each group. A modified version of a previously published protocol was used [44]. The DAM tubes were filled with the same food used to house all flies used for our studies. The food was melted in a beaker; sterilized DAM tubes were placed in the liquid food. After the food was given time to solidify, the tubes were removed from the beakers and the outsides were cleaned. Melted wax was used to seal the ends with the food to avoid drying out. Flies were placed onto a CO_2_ pad to anesthetize them, and one fly was loaded into an individual tube. A small piece of cotton was used to stopper the tubes with the flies. The tubes were loaded into the activity monitors and placed within an empty incubator maintained at 25 °C and 50% humidity. The monitors were connected to the computer as described in the cited protocol with equipment purchased from TriKinetics [44]. Data was collected over the course of 10 days with three days of a 12 h light:dark (LD) cycle, for entrainment, and 7 days of constant dark, dark:dark (DD), conditions. Data was summarized using the ShinyR DAM data analysis application [45]. The main analyses used in this study included actograms, activity in the day and night, and percentage of rhythmic flies in the 24 h dark conditions. Daytime versus nighttime activity changes were analyzed using data collected during the LD phase and comparisons between daytime and nighttime activity were analyzed via *t*-tests using GraphPad Prism (Version 10.0.0, GraphPad, San Diego, CA, USA). The percentage of rhythmic flies were evaluated using summary data generated by the periodogram feature of the ShinyR application. Differences in percentages of rhythmic flies between genotypes and treatments were statistically analyzed in GraphPad Prism (Version 10.0.0, GraphPad, San Diego, CA, USA) using a chi-square test. The column data sheet format in prism was used to generate a final graph of the percentage of flies that were considered rhythmic in each group, with ‘1′ being designated as a rhythmic fly and ‘0′ being designated as an arrhythmic fly.

### 2.8. Quantitative Reverse Transcriptase PCR

Flies were fed octopamine or ddH_2_O containing food for five days and then frozen on the morning of the sixth day, 1–2 h after lights on. Three independent biological replicates were generated for each genotype and treatment group. Total RNA was extracted from frozen whole flies using TRIZOL (Invitrogen, Waltham, MA, USA). The Power SYBR Green Master Mix (Applied Biosystems, Waltham, MA, USA) and the StudioQuant 3 Real-time PCR System (Thermo Fisher Scientific, Waltham, MA, USA) were used to perform one-step qRT-PCR. Twenty (20) μL reactions were generated using the extracted RNA (4 μL at 25 ng/μL), forward and reverse primers (1 μL each at 1 μM), Power SYBR Green PCR Master Mix (10 μL), reverse transcriptase (0.1 μL), RNAse inhibitor (0.025 μL), and ddH_2_O (3.875 μL). Three technical replicates were loaded onto a 96-well plate for each biological replicate. The qPCR program was set up to run for 40 cycles of the following: 95 °C for 15 s followed by 60 °C for 1 min. The mRNA data was normalized to Act5C. All flies used in qrtPCR analysis were males.

Primers used:

Act5C: F, 5′-CGCAGAGCAAGCGTGGTA-3′; R, 5′-GTGCCACACGCAGCTCAT-3′

Clock: F, 5′-TGGAGTCTCTCGATGGTTTTA-3′; R, 5′-CGGTGTGGGATTCATAAAGAT-3′

Sestrin: F, 5′-ATGTACTACGCCGTCGATTACT-3′; R, 5′-TCGTCCATGTCAAAGTCGGAT-3′

Spargel: F, 5′-GGATTCACGAATGCTAAATGTGTTCC-3′; R, 5′-GATGGGTAGGATGCCGCTCAG-3′

## 3. Results

### 3.1. Clk^out^ Mutants Have Severe and Robust Phenotypes in Climbing Speed, Endurance, and Flight Performance

We first evaluated mobility associated phenotypes of several circadian clock mutants obtained from the Bloomington *Drosophila* Stock Center (BDSC). Three of the mutants were in the *Canton-S* background and were compared to *Canton-S* controls [46]. All three, *per*^01^, *tim*^01^, and *Clk^jrk^*, showed significant reductions in endurance. Endurance was evaluated both early on in life, at five days of age, and later in life, at 28 days of age (Appendix A). The *tim*^01^ mutants also showed the most dramatic decline in endurance with age (Appendix A). However, the *per*^01^ and *Clk^jrk^* mutants did not show any significant climbing speed phenotypes and *tim*^01^ only showed a significant reduction in climbing speed later in life (Appendix A).

In comparison, *Clk^out^* mutants, generated in the *w*^1118^ background, had more diverse phenotypes and consistently performed worse than controls in all assessments performed. These mutants were also obtained from the BDSC and were raised and maintained on the same food media as the mutants in the *Canton-S* background [8]. They were confirmed to have very low *clock* expression when compared to controls (Appendix A). We also confirmed that in our hands the mutants lacked a circadian rhythm through the Drosophila Activity Monitor assay (Appendix A). Furthermore, *Clk^out^* mutants had significantly lower endurance than *w*^1118^ controls (Figure 1A). 

The mutants also had a lower climbing speed than controls during all three weeks of the climbing speed assessment, beginning at five days of age (Figure 1B). Because of these more severe and consistent phenotypes, we focused on the *Clk^out^* mutant for further experimentation. The flight performance of these mutants was evaluated, and the mutants were found to have worse flight performance than controls at seven days of age (Figure 1C). We also evaluated the performance of females in the same assessments. Female mutants, like males, were found to have significantly worse endurance when compared to *w*^1118^ controls (Figure 2A) and were found to have reduced climbing speed and flight performance, similar to males (Figure 2B,C).

Work from our lab and others has consistently shown improvements in health span with exercise training in *Drosophila* [34,47]. Endurance exercise has also been demonstrated to improve health span in mammalian models of rhythm disruption [6]. So, our next step was to test the effects of three weeks of exercise training on the *Clk^out^* mutants. Exercise training did not lead to any improvement of endurance, in the *Clk^out^* mutant animals regardless of sex (Figure 1D and Figure 2D). Female controls (*w*^1118^) also did not show significant improvements in run span after three weeks of training, as previously observed [38].

### 3.2. Activation of Exercise Response Pathways by Octopamine Can Rescue Exercise Phenotypes of Clk^out^ Mutants

The failure of a chronic exercise program to rescue the mutants’ endurance and climbing speed could be because exercise is not sufficient to rescue these phenotypes, but alternately, could be because the reduced exercise performance of the mutants prevented them from effectively completing the exercise program [6,24,25,26,27]. To distinguish between these possibilities, we decided to try activating exercise response pathways exogenously without the need for physical training. Precedents exist of *Drosophila* disease models with severe mobility deficiencies that could be effectively rescued by exogenous applications of exercise-mimicking molecules, i.e., exercise mimetics [42,48]. We have previously identified several such exercise mimetics [37,38]. We hypothesized that the observed *Clk^out^* phenotypes could be rescued by using exercise mimetics to bypass the need for physical training. 

We chose to rescue through the feeding of octopamine (OA), an invertebrate norepinephrine equivalent. We have previously shown that wild-type females do not respond to exercise training, and this difference is entirely due to sex differences in the activity of OA-ergic neurons during training [38]. OA release is critical for the response to chronic training, and the feeding of OA alone is able to mimic the effects of endurance exercise in both male and female wild-types [38]. As previously observed, in both male and female *w*^1118^ flies, OA feeding mimicked the improvements in run span induced by exercise training (Figure 3A,C). Furthermore, OA feeding was able to rescue the endurance of *Clk^out^* mutants, leading to significant improvements in run span regardless of training status (Figure 3B), while vehicle-fed controls retained the defects as in Figure 1. The results in female mutant flies were similar to males (Figure 3D).

OA feeding also led to significant improvements in the climbing speed of both *w*^1118^ and *Clk^out^* flies. Wild-type control males improved climbing speed with either training or OA feeding, with no additive effect between the two treatments (Figure 4A,C), similar to previous observations [38]. OA feeding caused significant improvements in the climbing speed of *Clk^out^* mutants, (Figure 4B,D) in contrast to exercise training, which did not improve the mutants’ climbing speed, regardless of sex (Figure 4B,D). However, the flight performance phenotype of *Clk^out^* was not rescued by OA feeding (Appendix A). 

To ask whether rescue of performance by OA acted by rescuing circadian rhythms, or acted independently of the mutants’ circadian defects, we measured activity rhythms during OA feeding. OA feeding did not significantly impact the activity rhythm of either the *Clk^out^* flies or the *w*^1118^ wild-type controls (Figure 5). OA-fed mutants still had fewer flies with a circadian rhythm than controls (Figure 5A). Mutant males still had slightly higher activity levels when compared to controls and still showed no reductions in activity during the dark (night) hours of the DAM assessment. Mutant females also showed no change in activity during the dark hours, whereas the control females had an obvious reduction in activity during this period. The only exception was the mutant females that had been fed OA, which did show a trend toward reduced activity during the dark period, although it was not statistically significant (Figure 5B). 

### 3.3. Octopamine Feeding Leads to Increased Expression of Two Exercise Response Transcripts

To further elucidate the mechanism by which octopamine feeding rescues endurance and speed of *Clk^out^* mutants, we measured changes in gene expression of exercise-related genes. First, we confirmed that *Clk^out^* mutants had a reduced expression of *Clk* at baseline and that OA feeding did not increase *Clk* expression in the mutants when compared to *w*^1118^ controls (Figure 6A,B), as expected given the amorphic nature of the *Clk^out^* mutation [8]. We also measured expression two more genes, chosen because both have previously been shown to increase in response to exercise training. These included the PGC-1α homolog Spargel and the stress inducible protein Sestrin. Sestrin has been shown to be critical for exercise training to increase speed and endurance [37,42]. Spargel is regulated by Sestrin and has been shown to be critical for Sestrin’s effects in the context of exercise training [37]. We found that at baseline, *Clk^out^* mutants had a reduced expression of both *spargel* and *sestrin* when compared to controls (Appendix A). After OA feeding, the expression of *spargel* and *sestrin* was increased in both control and mutant flies (Figure 6C,D). Furthermore, following OA feeding, the mutants increased expression of both exercise-mediating genes even more than controls did (Appendix A), perhaps in compensation for their lower starting levels. 

## 4. Discussion

Here, we described defects in exercise performance of a circadian clock mutant, *Clk^out^*, including an inability to respond to exercise training by improving speed and endurance (Figure 1 and Figure 2). We have previously demonstrated that the exercise training protocol used in this study leads to improvements in endurance, flight performance, and climbing speed in wild-type flies [41,47]. Because mutants have reduced speed, it is possible that they take fewer “steps” during the training program, and this may contribute to the lack of response to training. Furthermore, the mutants were found to have reduced baseline expression of two genes that are required for the exercise response, *sestrin* and *spargel*, which also likely blunts their ability to respond to training (Figure 6). 

Sestrins are evolutionarily conserved, stress-inducible proteins that are critical for the exercise response in both invertebrates and mammals [37]. *Drosophila* Sestrin has been shown to act through multiple pathways to induce an exercise response and one of these pathways includes the upregulation of Spargel [37,42]. Spargel has been shown to be important for the exercise response in both flies and mammals and its homolog is also a known modulator of clock gene expression in mammalian studies [49,50]. These results are consistent with other models of circadian disruption that have reduced expression of these exercise-related proteins [51,52]. This reduced exercise capacity, along with reduced levels of molecules required for an exercise response may explain why we, consistent with others [53], did not find that exercise could effectively rescue the effects of rhythm disruptions.

Reduced baseline exercise capacity has also been seen in other models of circadian disturbance [6,12,28]. Mice with central clock mutations demonstrate a reduced ability to exercise, assessed through a moderate intensity treadmill test [6,28]. Studies in humans, specifically shift workers, also demonstrate reduced exercise levels although this is usually the result of time constraints that come with life events which produce circadian disruptions [12]. These events often prevent people from achieving exercise at the optimal time of day and receiving the full benefits of exercise [12,28]. Despite this evidence of reduced exercise capacity or access, it has been demonstrated that exercise can still be beneficial in the context of rhythm disruption [6,23]. Mouse models of circadian disruption have improved rhythms of body temperature, activity, and heart rate following voluntary exercise [6]. Exercise has also been demonstrated to improve metabolic, cardiac, and neurological health both within and outside the context of rhythm disruption [21,22,23].

One way to bring the benefits of exercise training to shift workers, with a reduced access to exercise, could be to stimulate downstream molecules induced by exercise that mediate its benefits [37,38]. Here, we demonstrate that such an approach can indeed work in a small animal model, using the exercise-induced mimetic, octopamine (OA) [38].

OA was able to fully rescue the reduced endurance phenotypes of the *Clk^out^* mutant to the level of the wild-type controls (Figure 3), and partially rescue climbing speed (Figure 4), without rescuing flight performance (Appendix A). These results suggest that OA is more effective at restoring long-term, energy consuming activities in *Clk^out^* mutants than at restoring short-term activities requiring a quick burst of energy, such as flight induction. The mechanistic reasons for this are unclear, given that OA can increase all of these performance characteristics in wild-type flies. OA also increased the expression levels of at least two key exercise response genes, which likely mediate the rescue effect of OA. Further work will be necessary to understand whether *sestrin* and *spargel* are required for the rescuing effects of OA in *Clk^out^* mutants, or whether overexpression of *sestrin* or *spargel* alone could also produce a similar rescue. 

Octopamine also serves as a wake-promoting neurotransmitter that regulates the circadian rhythm [54], leading us to ask if OA feeding could act, in part, by stabilizing the activity rhythms of *Clk^out^* mutants. However, in our hands, OA feeding did not have much impact on the activity rhythms of either mutants or wild-type flies (Figure 5). Another potential explanation for the rescue produced by octopamine feeding is that exogenous octopamine may act on peripheral clocks, like those in the muscle, to rescue peripheral rhythms that may be involved in the metabolic changes required in muscle to produce benefits from exercise [55]. Release of myokines from muscle has been shown to be important for the metabolic changes that occur in other body tissues to allow for adaptations to exercise, and *Drosophila* OA receptors have been shown to be required for exercise-induced changes in other tissues, suggesting OA must regulate secondary messengers emanating from muscle [56]. Further supporting this idea, OA has recently been shown to upregulate expression of a putative myokine in flies, the FNDC5/Irisin homolog *Iditarod* [57].

A recent study has identified another potential mechanistic target of exercise in this context, a micro-RNA, *dmir283*, that may also be involved in the beneficial effect of exercise on aging circadian rhythms [58]. While our study focuses on a circadian mutant rather than age-related decline of circadian rhythms, this micro-RNA may be another avenue to investigate the mechanisms behind the rescues demonstrated here. 

## 5. Conclusions

Here, we show that a *Drosophila* circadian mutant has reduced exercise performance, and is resistant to improvement by involuntary exercise training. We also show that this reduced exercise capacity can be rescued by an upstream molecule known to mimic exercise in other contexts (Table 1 and Table 2). Understanding the mechanisms behind the beneficial effects of exercise and exercise mimetics on circadian rhythm disruption may reveal novel therapeutic strategies to provide the benefits of training to those unable to exercise or unable to receive the full benefits of endurance exercise, whether due to time constraints or the reduced exercise capacity associated with circadian disturbance.

## Figures and Tables

**Figure 1 cells-12-02515-f001:**
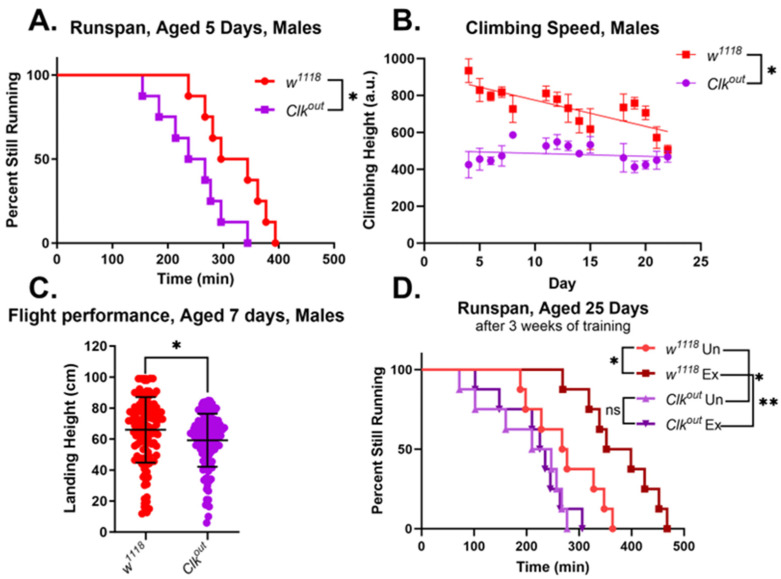
***Clk^out^* mutants have impaired performance in multiple mobility associated assessments**. *Clk^out^* mutant males showed reduced endurance even at five days of age compared to background controls (**A**). Each data point depicts time to exhaustion for 80% of flies in an individual vial (*n* = 8 vials of 20 flies each) analyzed via log rank. *Clk^out^* mutants also have reduced climbing speed across the first three weeks of life (**B**). Each data point represents the average climbing speed over four pictures taken daily (*n* = 5 vials of 20 flies each) analyzed longitudinally via a two-way ANOVA. The trend line represents the slope of linear regression. The flight performance of *Clk^out^* mutants is also reduced (**C**). Average landing height between the two groups (*n* = 8 vials of 20 flies each) was compared using a *t*-test. *Clk^out^* mutants showed no improvement in endurance following three weeks of chronic exercise training while the control *w*^1118^ flies showed a significant improvement, *n* = 8 vials of 20 flies each (**D**). * *p* < 0.05, ** *p* < 0.001, ns is not significant.

**Figure 2 cells-12-02515-f002:**
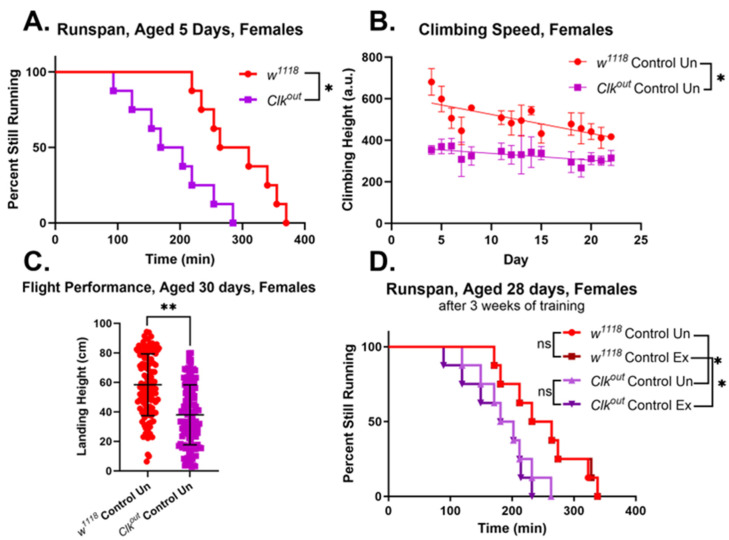
***Clk^out^* mutant females have impaired exercise performance.** At 5 days of age *Clk^out^* females had significantly lower endurance than *w*^1118^ controls, *n* = 8 vials of 20 flies each, (log-rank) (**A**). *Clk^out^* females also had significantly reduced climbing speed, *n* = 5 vials of 20 flies each, (two-way ANOVA), line represents linear regression (**B**). Flight performance in *Clk^out^* females was also significantly worse than controls, *n* = 8 vials of 20 flies each, (*t*-test) (**C**). Neither *Clk^out^* females, nor *w*^1118^ females showed any improvement in endurance with three weeks of endurance exercise training, *n* = 8 vials of 20 flies each, (log-rank analysis) (**D**). Control data displayed here, to allow for direct comparisons between untreated mutant and untreated control female flies, is also shown in rescue experiments in later figures. * *p* < 0.05, ** *p* < 0.001, ns is not significant.

**Figure 3 cells-12-02515-f003:**
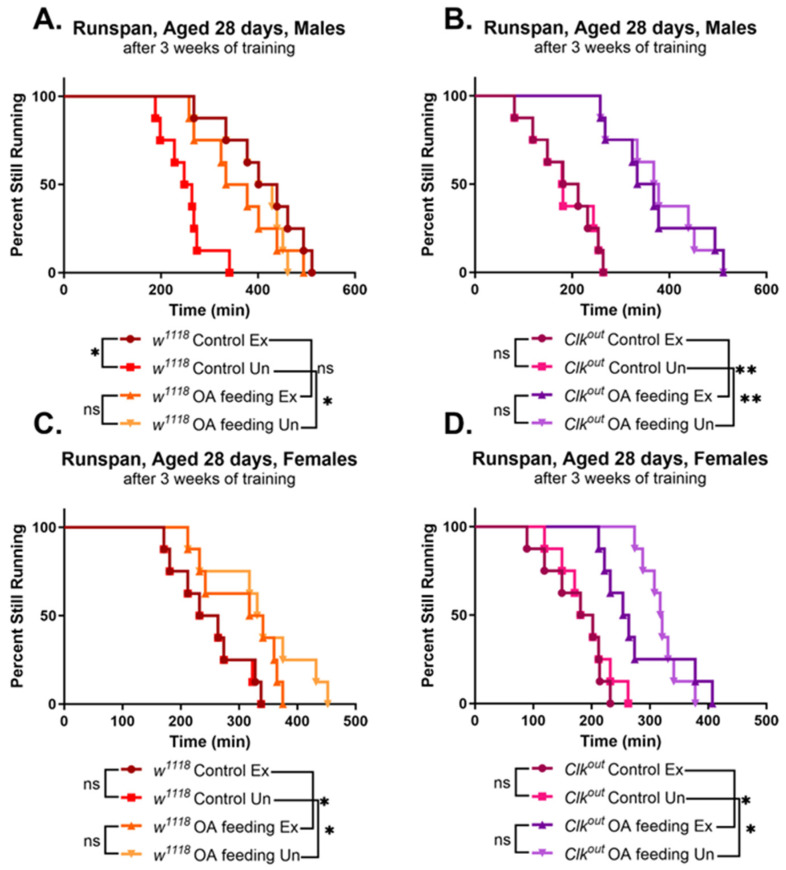
**Octopamine feeding rescues the endurance phenotype of male and female *Clk^out^* mutants.** Male controls responded to exercise training (**A**), while females did not (**C**), as expected. Three weeks of OA feeding led to significantly improved endurance in male (**A**) and female (**C**) control flies and had no additive benefit with exercise. OA feeding rescued the endurance of *Clk^out^* males (**B**) and females (**D**) regardless of training status. *n* = 8 vials of 20 flies each, log-rank analysis. * *p* < 0.05, ** *p* < 0.001, ns is not significant.

**Figure 4 cells-12-02515-f004:**
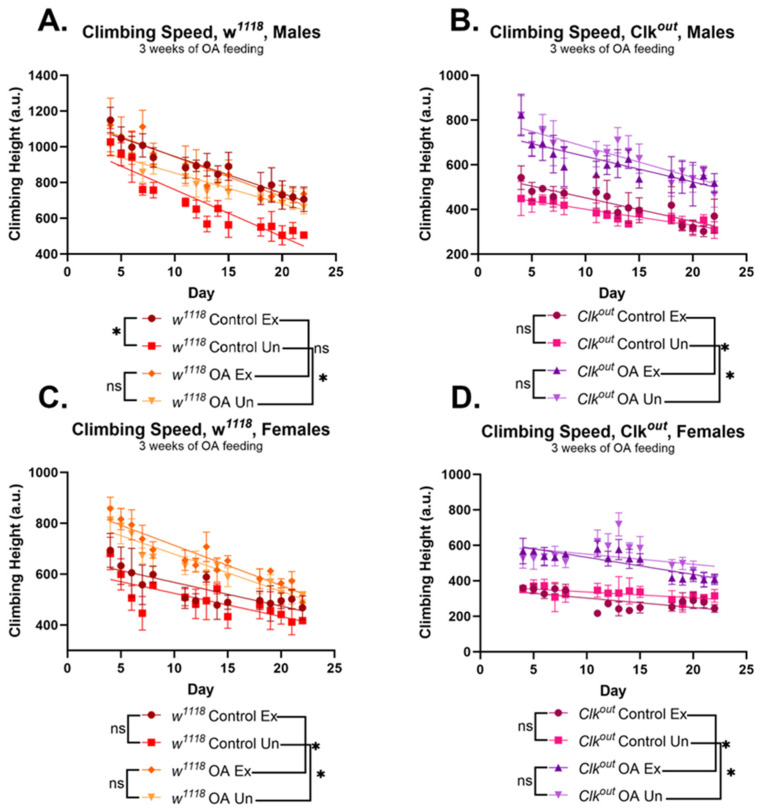
**Octopamine feeding rescues the climbing speed phenotype of male and female *Clk^out^* mutants.** Three weeks of octopamine (OA) feeding improved climbing speed of *Clk^out^* mutants, while exercise training provided no additive benefits. Exercise training did not rescue climbing of *Clk^out^* mutants in the absence of OA. Similar results were seen in both male and female mutants (**B**,**D**). *w*^1118^ male controls showed improved climbing speed in response to exercise training or to OA feeding, and training did not have additive benefits with OA feeding in males (**A**). *w*^1118^ females did not show any improvements in response to training but did show significant improvements with OA feeding, consistent with previous results [38] (**C**). Age had a significant effect on climbing speed in all groups (*p* < 0.0001 for main effect of age) and had a significant interaction with exercise in *w*^1118^ males (*p* = 0.0053), and a significant interaction with OA feeding in all but *w*^1118^ males (*Clk^out^* males *p* = 0.002, *Clk^out^* females *p* < 0.0001, *w*^1118^ females *p* < 0.0001). *n* = 5 vials of 20 flies each, two-way ANOVA and three-way ANOVA, lines represent linear regression. * *p* < 0.05, ns is not significant.

**Figure 5 cells-12-02515-f005:**
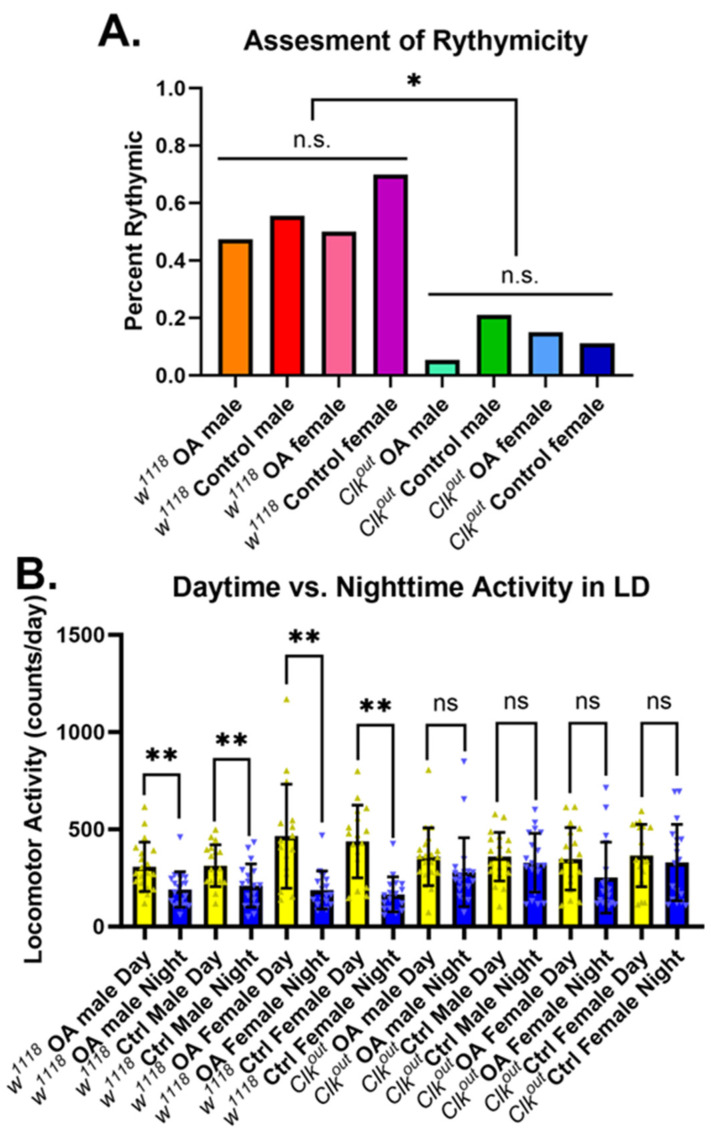
**Octopamine feeding does not rescue activity rhythm phenotypes of *Clk^out^* mutants**. No significant changes were observed in the percentage of rhythmic flies within the control or experimental groups after OA feeding. However, all the mutants performed significantly worse than the controls. All comparisons under the bars (control vs. OA) were not significant while all those between bars (*w*^1118^ vs. *Clk^out^*) were significantly different, *n* = 16 (**A**). Main effects of OA treatment and genotype were analyzed via two-way ANOVAs; genotype was found to have a significant main effect (*p* = 0.0150) while OA feeding did not have a significant main effect (*p* = 0.1337). Percent rhythmicity between groups was compared via a chi-square test. Regardless of treatment or sex, all control flies had significantly lower activity during the nighttime hours. The mutant cohorts showed no significant decrease in activity during the nighttime hours. The only possible exception were mutant females which trended toward a lower activity in the dark hours; however, the difference in activity from light to dark hours was not significant (*p* = 0.1014). *n* = 16, *t*-test (**B**). * *p* < 0.05, ** *p* < 0.001, ns is not significant. The assessment of rhythmicity was conducted in DD conditions, while daytime vs. nighttime activity was assessed in LD.

**Figure 6 cells-12-02515-f006:**
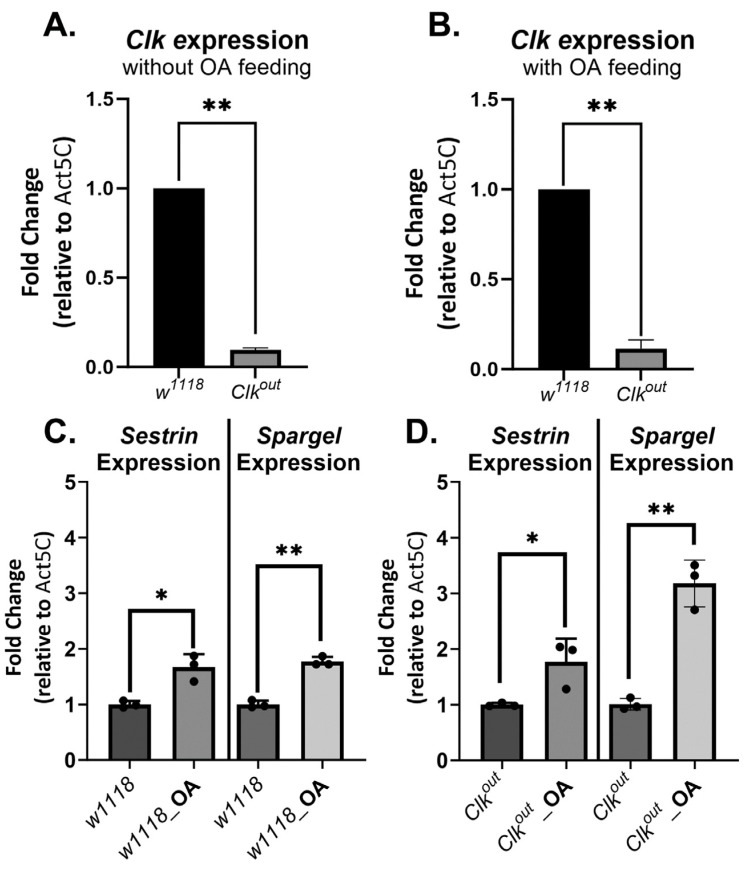
**Octopamine feeding increases expression of exercise response genes in *Clk^out^* mutants but does not impact *Clock* gene expression.** qRT-PCR was used to evaluate expression of key genes in mutants and controls. *Clk^out^* mutants had lower *clock* gene expression when compared to *w*^1118^ controls, *n* = 3 biological replicates, (t-test) (**A**). OA feeding did not cause a significant change in the expression of *clock* in the mutants (**B**). However, OA feeding did lead to increased expression of both exercise response genes, *sestrin* and *spargel*, in both *w*^1118^ (**C**) and *Clk^out^* (**D**). The delta Ct method was used to compare gene expression between flies fed OA and those fed normal food, *n* = 3 biological replicates with three technical replicates within each biological replicate, *t*-test, * *p* < 0.05, ** *p* < 0.001.

**Table 1 cells-12-02515-t001:** Summary of endurance data.

Endurance				
	*w*^1118^ Males	*w*^1118^ Females	*Clk^out^* Males	*Clk^out^* Females
Exercise	+	-	-	-
OA Feeding	+	+	+	+
Exercise x OA	-	-	-	-

**Table 2 cells-12-02515-t002:** Summary of climbing speed data.

Climbing Speed				
	*w*^1118^ Males	*w*^1118^ Females	*Clk^out^* Males	*Clk^out^* Females
Exercise	+	-	-	-
OA Feeding	+	+	+	+
Exercise x OA	-	-	-	-

## Data Availability

The data presented in this study are available in Appendix A.

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
