# Peer review of "Octopamine Rescues Endurance and Climbing Speed in Drosophila Clkout Mutants with Circadian Rhythm Disruption"

_cells, 2023, doi:10.3390/cells12212515_

Round 1
Reviewer 1 Report
In this manuscript, the authors address the complex interplay between genetics, aging, exercise and pharmacology in the Drosophila model. The studies address an interesting and important biological problem with clear implications for human health. The complex nature of the interplay between the various facets of their study, while unavoidable, does tend to make the data and interpretations somewhat challenging to follow unfortunately. That intrinsic challenge aside, the authors are experts in their field and, unsurprisingly, all experiments and statistical analyses appear to be performed with the necessary level of attention and thoroughness. Overall, the manuscript advances the field by showing that Clk mutants have a variety of behavioral phenotypes, the invertebrate neurotransmitter octopamine improves the locomotor performance of Clk mutants without altering their rhythmicity or response to excercise. I have only relatively minor comments and suggestions for improving the manuscript:
1. I think the text lines 253-288 is difficult to follow and I think a slight re-ordering of the results might be helpful. My suggestion is to show and describe the control w1118 data in panels 3A/C first then show and describe the Clk data in panels 3B/D second. My suggestion is to similarly re-order the data in figure 4 so the w1118 data are shown/described first followed by the Clk data. One final suggestion is to explicitly state whether age, genotype, octopamine treatment and exercise affect performance in each case and whether there are interactions between these factors. A table that summarizes the findings (affects vs no affects) might be warranted given the intrinsic complexities involved.
2. The feeding of octopamine has a rather dramatic effect. In the data as shown, however, it is not immediately clear to me whether those effects are due to octopamine consumption per se, altered consumption of dietary media laced with octopamine, or a combination of the two. This could be sorted out by explicitly measuring the consumption of food media using any of the methods in the articles below. My suggestion is for the authors to perform targeted experiments to measure consumption of octopamine-treated food, to cite a previous demonstration of octopamine consumption, or to state in the text that alternative interpretations are possible in the absence of food consumption measurements.
A. Park, T. Tran and N. S. Atkinson. Proc Natl Acad Sci U S A 2018 Vol. 115 Issue 36 Pages 9020-9025
S. A. Deshpande, G. B. Carvalho, A. Amador, A. M. Phillips, S. Hoxha, K. J. Lizotte, et al. Nat Methods 2014 Vol. 11 Issue 5 Pages 535-40
B. C. Shell, Y. Luo, S. Pletcher and M. Grotewiel. Sci Rep 2021 Vol. 11 Issue 1 Pages 20044
Q. Wu, G. Yu, S. J. Park, Y. Gao, W. W. Ja and M. Yang. iScience 2020 Vol. 23 Issue 1 Pages 100776
No additional comments.
Reviewer 2 Report
Summary: The authors performed experiments to determine if exercise and/or octopamine (and exercise mimetic) was sufficient to restore exercise capacity and circadian rhythmicity in flies (drosophila) with a genetic deletion of a core circadian clock gene (Clkout). They found that exercise restored some of the exercise deficits (mostly in males), and OA produced similar effects. These data suggest that exercise mimetics may be useful to provide exercise-like stimuli to those with disrupted circadian rhythms, like shiftworkers. Several major and minor comments are below.
Major Comment:
- Exercise Implementation: It would be beneficial to report the timing of exercise stimulus with respect to circadian rhythm of flies, at a minimum to active/rest phase, perhaps to a ZT window.
- Circadian rhythm and Activity
o Is it more appropriate to call this ‘activity rhythm phenotype’ vs ‘sleep’?
o Can you show non-binary results of circadian rhythm assessment? i.e. – Activity Acrophase?
o It should be stated in the figure legend that the rhythmicity assay was performed in DD, while the daytime/nighttime activity assessment was perform in LD (as seen in methods).
o Also, it seems like OA reduced the percentage of rhythmic flies in 3 of the 4 comparisons. Was there a statistical effect of OA (main effect)?
o The comparisons made in text (Line 304) are not well presented in the figure in regards to group effects (independent of LD cycle). Could warrant an additional figure for total activity, or additional statistical markers for group wise comparisons.
- Figure 6: Very difficult to interpret. 6A and 6B should likely be combined into one figure, especially if comparison between the control groups with and without OA feeding are to be made (which appear to be arbitrarily normalized to 1.0, with no error). The labeling of 6C-D must be improved for clarity. I’m guessing based on the legend that the first two bars or Sestrin expression in w1118 flies on control and OA diet, respectively? Are the male or female flies? Also, comparison of Sestrin and Spargel between w1118 and Clkout flies (line 338 and 356) cannot be made since they are analyzed separately (and normalized to 1.0 in control groups).
o Typically, performing qRT-PCR allows for the quantification of transcripts, which don’t need to be presented as fold change any more.
o If presenting data as fold change from a control group, it is still necessary to calculate that fold change in a way that presents the appropriate experimental error in the reference group (not 1.0 +/- 0.0 with no error bar).
§ To do this, you can divide all ddCt values by the [AVERAGE OF THE CONTROL GROUP], including all of the values in the control group. This give you a control group with an average of 1.0 and the real error around that mean.
- THROUGHOUT: be consistent with use of italics when referring to genes, subscript/superscript when referring to flies, etc.
- Since the OA is provided by ad libitum feeding, what is the eating rhythm like, food intake, etc? How long does ingested OA remain bioactive? I think this is critical to address if we are to infer how this compound alters/improves circadian rhythmicity.
Minor Comment:
- Line 60: mice are nocturnal
- A better description of Clkout mice would be warranted even though they are commercially available.
- Line 135: turkey’s should be Tukey’s
- Methods/Flight Performance: difficult to interpret. Not sufficient detail in the methods to replicate
- Line 147: put DAM in the parentheses to define the abbreviation
- Consistently use Clkout, Clkout, etc throughout
- Line 256: insufficient should be sufficient
- Line 387: allele? Stay consistent with the rest of the manuscript and call them Clkout flies
- Line 414: miRNA or micro RNA
- I think the discussion struggles to differentiate between behavioral difficulties (i.e. – can’t find the time to exercise) and mechanistic inhibition (exercise does not produce benefits even if exercise is performed) of exercise effects in shift working population, or present flies. This could potentially be addressed by quantifying the exercise dose in this study between genotypes, and explicitly describing this as chronic ‘voluntary’ or ‘forced’ exercise training. If it is forced training at an equivalent dose, then it is more applicable to the mechanism.
Author Response
please see attached word doc

Reviewer 3 Report
1. Supplementary figures are missing. The supplementary files included only had the data deposit files, with no figures.
2. In the interest of transparency, all data points should be shown in bar graphs (Figures 5 and 6).
3. Authors should state in the methods at what time (or zeitgeber time) samples were collected for experiments.
4. The experiments in Figure 6 have significantly less replicates than the rest of the experiments. Additionally, the data points are not shown. Are the authors appropriately powered in these experiments?
5. The colors the authors use for the figures are difficult for colorblind individuals to see. The authors should consider using colorblind-friendly color pallets.
6. Are there known differences between the W1118 and Canton-S backgrounds that could explain differences in phenotypes seen in the different KO groups? Do female Canton-S show increased run span with exercise training like the W1118 male controls? If it is known that the W1118 background females do not respond to exercise training, is this the best model for studying the effects of Octopamine on females? Or at minimum, this may mean there are different mechanisms for the effect of Octopamine on females vs. males. This is a limitation of the study which should be discussed.
7. Line 272-273: Exercise training had no effect on run span in female w1118 flies.
Author Response
please see attached word doc

Round 2
Reviewer 2 Report
I still believe that the figure is not appropriately labeled in the figure 6C and 6D. The bars should be labeled by biological group (i.e. - Control, OA fed), not Control, Spargel, for example. Spargel expression was measured in both bars, so the figure should be labeled accordingly.Author Response
We agree with the reviewer's point and we have altered the figure to have the transcript labelled at the top and the genotype and feeding state below each bar. We hope this is much clearer than the previous version and thank the reviewer for pointing out the issue
Reviewer 3 Report
The authors addressed all of my concerns.
Author Response
We thank the reviewer for their helpful comments